# Extractive Fishing Gear in the Mazarrón Bay (Murcia Region, Spain) during the First Half of the 20th Century: A Heritage Prone to Being Forgotten

Daniel Moreno-Muñoz [1], Ramón García-Marín [2,*] and Cayetano Espejo-Marín [2]

1   Department of Social Science, Philosophy, Geography and Translation and Interpretation, University of Córdoba, 14003 Córdoba, Spain; gt2momud@uco.es
2   Department of Geography, University of Murcia, 30001 Murcia, Spain; cespejo@um.es
*   Correspondence: ramongm@um.es; Tel.: +34-868-88-31-46

**Abstract:** Fishing in the Mazarrón Bay has been practiced since prehistoric times. This was one of the basic pillars of the area's economy; however, due to the development of tourism, this maritime activity has been pushed into the background. The changes in the fishermen's way of fishing in the last decades of the 20th century, as a consequence of the proliferation of boats with greater extractive capacity, have meant that much of the fishing gear has fallen into disuse. The main objective of this research is the compilation of information on their use and the available tools of this heritage in order to preserve their history. In order to do so, the archives of the Mazarrón Fishermen's Guild were consulted and interviews were conducted with local fishermen over 80 years of age, who fished with devices that are no longer in use today. The results show that it is possible to promote them as tourist attractions, with the aim of raising awareness of the fishing identity and the environment in which it is practiced, in the southeast of the Iberian Peninsula.

**Keywords:** fishing arts; fishing cultural heritage; Mazarrón Bay; extractive fishing





## 1. Introduction

Fishing is the extractive activity that provides humans with food resources that exist in marine and inland waters [1]. In other words, along with hunting and gathering, it is one of the oldest activities in human history [2]. It is practiced using all kinds of devices, gadgets and techniques to catch fish and has become a way of life for the populations of coastal territories [3], a situation that has left a cultural richness in the places where it is practiced, as it plays an essential role in the food and income of societies in many parts of the world [4].

Artisanal fishing in Mediterranean countries has historically played a decisive socio-economic role in coastal areas [5–7] and one of its most important characteristics is the use of a variety of fishing gear that allow different species to be caught in a sustainable manner [8]. In recent decades, fishing has evolved considerably due to technological innovations in vessels that allow a greater extractive capacity of marine resources [9], also leading to sustainability problems in fishing grounds [10,11]. This fact has meant that traditional fishing gear is no longer used for fishing, one of the measures introduced by national and international policies governing the fishing sector [3], which in parallel entails a loss in terms of fishing cultural heritage [12] and is also ceasing to form part of the collective memory of local populations [12].

Fishing is, in itself, an experiential activity, in contact with an ecosystem such as the marine one, full of suggestions and nuances; an experience that can arouse great interest in society as a result of its natural, fishing and cultural resources (lighthouses, ports, fish markets, traditional fishing systems and gear, contact with the people of the sea and their own vocabulary, a set of intense festive events associated with the sea as an inspirational

framework and as a setting in which to take place, its museums and interpretation centers, gastronomy, monuments and crafts). In other words, fishing communities are places characterized by the human activities and social processes that have taken place there [13,14]. The so-called fishing societies have lived with a clearly differentiated cultural identity marked by the fact that many cultural elements were, to a large extent, influenced by fishing and the sea [15]. Fishing goes beyond its merely economic function, as a means of subsistence, to influence all areas of culture: ways of speaking about the place, landscapes, festive expressions, beliefs, food, clothing, lexicon, etc. In this sense, it can be argued that fishing and its culture are part of a maritime cultural landscape and a traditional source of labor [16]. These places, through dissemination actions organized by the local population or the authorities, help to understand the culture of fishermen and the significance of heritage in their daily lives [17].

The concept of heritage implies that the tangible and intangible goods that form it have been inherited by a social group and preserved over time to be passed on to future generations. In other words, in coastal territories where fishing is or has been an important activity, we can speak of the existence of a broad fishing cultural heritage, both tangible and intangible. Nevertheless, as a result of the tourist development that began in the second half of the 20th century, many areas that were once small fishing villages have lost much of their cultural heritage linked to fishing activity [18]. Traditional fishing gear and its use have been disappearing due to the dynamics of the sector. For example, nowadays it is not fished with fishing gear such as *sardinal*, *andana*, *boliche* or the *jábega* that had a specific boat with its own characteristics to fish in places such as Málaga. According to [19], it is necessary to enhance the value of elements related to fishing activity and coastal areas, in order to prevent their deterioration and subsequent disappearance of these heritage elements. The protection and conservation of heritage helps define the sense of identity of a people and that a part of its history can be a source of social cohesion. In this sense, and aspects such as social alterations resulting from the changing lifestyle of the population [20], gentrification modifying the heritage of port cities or the intensification of mass coastal tourism [21] make the local population become more aware of heritage, demanding a greater protection of it, as well as its enhancement in the face of the loss of idiosyncrasy of the place.

In recent years, tourism linked to fishing, better known as seafaring or fishing tourism, has become one of the most important vectors for promoting the heritage linked to fisheries. According to [22], this type of tourism encompasses activities in fishing ports (gastronomic tastings, visits to auctions and port facilities, knot workshops or guided tours of heritage elements that are directly or indirectly related to fishing) and at sea (in situ viewing of the fishing day or recreation of traditional fishing gear). As a result, a great deal of work is being performed to enhance the value of fishing-maritime culture, favoring its conservation [23], and there are innovative proposals that respond to the need to diversify fishing and tourist activity [24] in order to achieve sustainable development. In this sense, the proliferation of fishing tourism activities is linked to European Fisheries Fund subsidies aimed at the economic diversification of coastal communities, as well as to regional policies that incorporate this form of tourism. The Fisheries Local Actions Group (FLAG's) are partnerships between fisheries actors and other local private and public stakeholders. Together, they design and implement a local development strategy to address their area's needs be they economic, social and/or environmental. Based on their strategy, the FLAG's select and provide funding to local projects that contribute to local development in their areas, involving thousands of local stakeholders. That is to saycan play a decisive role when it comes to valuing disused fishing gear, since these associations are the main promoters of the patrimonialization of maritime culture [25].

In the Mazarrón Bay (Region of Murcia, Spain), fishing has been present since prehistoric times, although its relevance in the late Roman period stands out as it was one of the main centers of salted fish production [26]. In recent times it has experienced a boom, especially in the last decades of the 20th century, with the incorporation of vessels of greater length and extractive capacity. Despite this, the traditional fishing gear gradually

disappeared from the sector, practically all of it falling into disuse. Nowadays, fishing activity continues to be very important in Mazarrón, but the sector is showing a regression due to the withdrawal of vessels and, therefore, to the loss of labor [27]. Given these circumstances, various organizations such as the Fishermen's Guild and the Town Council through the FLAG of Murcia Region (GALPEMUR) are looking for ways to promote the fishing culture in the municipality in order to preserve it as a unique competitive feature compared to other nearby tourist destinations [28].

In this sense, it has been considered appropriate and necessary to carry out an investigation in which we compile in written form the artisanal fishing gear and systems that were used in the Mazarrón Bay over a large part of the 20th century and how they were used by the fishermen. The main objective is to record them so that they are not forgotten or disappear from the collective imagination, as this is a material and immaterial fishing heritage. The oral accounts of the fishermen have also been gathered in order to preserve them with the possibility of making the most of them for tourism. In this way, fishing gear could be occasionally recreated on the coasts of the Mazarrón Bay as a way of enhancing maritime culture and oral accounts preserved in local museum. For this reason, the opinion of the fishermen on this matter and its possible tourist use have been collected, since in the event of carrying out an activity of this nature, it could mean the visit of people from other places to contemplate it and the local population would learn more about its past.

## 2. Materials and Methods

### 2.1. Area of Study

The Mazarrón Bay is located in the Region of Murcia (Spain), surrounded by Cabo Tiñoso and Cabo Cope, and has a coastal length of 45 km [29], being the deepest arc of the Gulf of Mazarrón, where another smaller geographical feature is located, the Ensenada or Rada de Mazarrón between Cabezo de los Aviones and Cabo Tiñoso [30,31] (Figure 1).

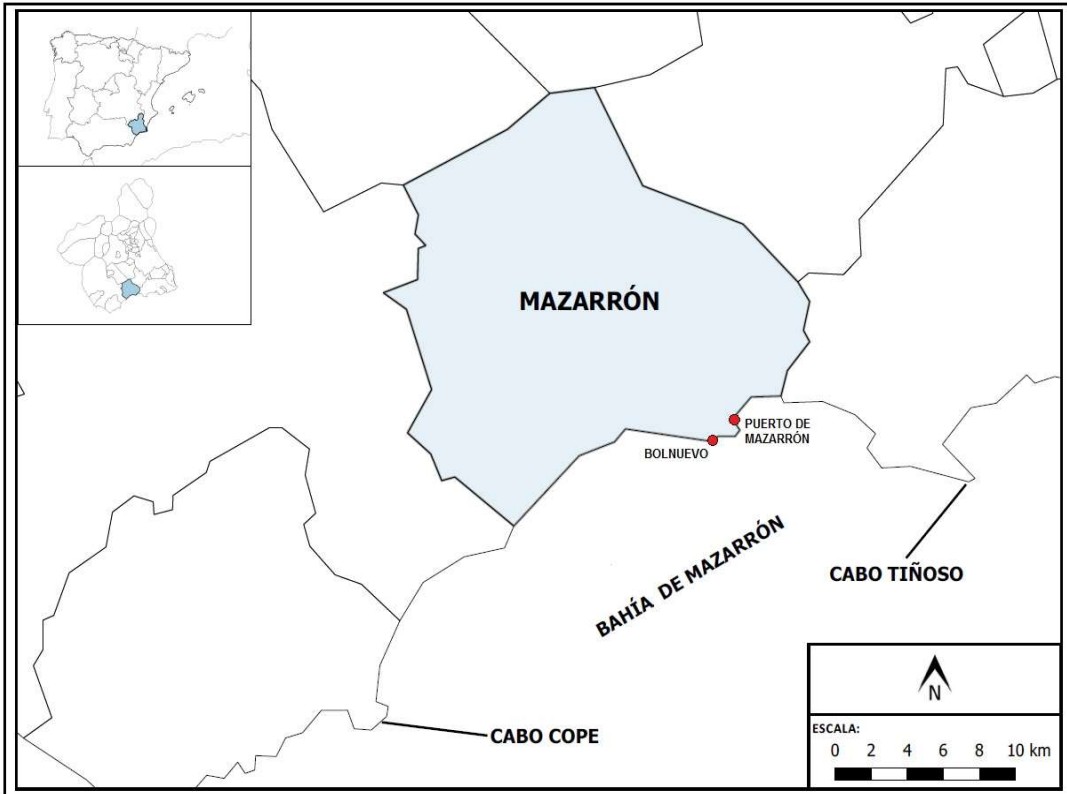

**Figure 1.** Geographical location of the Mazarrón Bay. Source: Prepared by the authors.

According to Köppen's climatic classification, this geographical area has a warm semi-arid climate (Bsh), in which rainfall is scarce (approximately 200 mm per year), concentrated in a few days, and the average annual temperature is 19 °C. The most influential elements for fishing activities are winds and waves. With regard to the former, the winds that predominate in the Mazarrón Bay come from the first quadrant (more than 30% per year), with the most common being the gregarious winds which come from the north east. On the other hand, the most common swell is from the east. The average annual height of the waves does not exceed 2 m, although when there are easterly storms they can exceed 4 m in height. However, this is not frequent, as the waters of this bay do not usually experience strong swells.

There was no fishing port in the Mazarrón Bay until the construction of the Puerto de Mazarrón port in the early 1970s. Until then, the fishermen's boats were grounded on the shores of the beaches of the towns of Puerto de Mazarrón and Bolnuevo or anchored close to the coast. Moreover, at the same time, houses for fishermen were built [13] in Puerto de Mazarrón, and some Bolnuevo fishermen changed their place of residence to larger houses with more amenities.

The fishermen fished during the first half of the 20th century, according to those interviewed, in areas of the Mazarrón Bay located a few miles from the places where the boats were based and at a short distance from each other. In other words, small-scale fishing was carried out, in which the volume of catches was much lower than it is today and, consequently, was characterized by a lower environmental impact on the marine environment.

*2.2. Methodology*

Conducting this study requires an appropriate methodological approach. There are two main ways of approaching research: one oriented towards measurements and the other towards experience, that is, one quantitative and the other qualitative, both enriching the understanding of the analyzed material [32]. The type of research developed in this study falls into the category of "non-experimental" and subcategories: descriptive, historical, correlational and qualitative [33].

There are several phases to the methodology carried out. In the first phase, a bibliographical review was carried out in order to have a more solid basis of knowledge about fishing cultural heritage and fishing activity in the Mazarrón Bay, in order to identify, evaluate and synthesize what has been carried out by other researchers. In addition, the archive of the Fishermen's Guild of Mazarrón has been consulted in depth in order to obtain information on the fishing gear and systems that were used in the territory that is the subject of this study during the first half of the 20th century. The reason that explains this time frame is the growth of the sector at the beginning of that century in Puerto de Mazarrón and Bolnuevo when fishing with new fishing systems [34].

In the second phase, a total of 20 interviews were carried out with retired fishermen who were over 80 years old, since they are the only people in Mazarrón who have fished in the first half of the 20th century. The interview consisted of 5 open questions: What fishing gear was used in the Mazarrón Bay in the mid-20th century, how was this gear handled, what species were caught with each of them, what were the fishing areas, and what is their opinion on the need to preserve them and the possibility of enhancing their value? In this sense, in the beginning, all of them fished in the sector with artisanal fishing gear but they fished according to the systems used by the shipowners depending on the time of year, the *jábegas*, *nasas* and *boliches* being the most mentioned by the interviewees. In addition, it is worth mentioning that once innovations arrived in the sector with boats with greater extractive capacity, artisanal fishing gradually disappeared, and the interviewees developed their professional careers in the purse seiner (13) and trawling (7) boats. In addition, 12 owners of small-scale fishing boats who are currently fishing were asked about the last question, since two kinds of fishing gear mentioned by the retired fishermen are still in use today. All the interviews were carried out in the facilities of the fishing port of

the Mazarrón Bay, as well as in the headquarters of the Fishermen's Guild, some of them in groups and others individually. Finally, once the information gathered had been compared, this study was drafted.

## 3. Fishing Culture in the Past in the Mazarrón Bay

Up until a few decades ago, the towns of the Mazarrón Bay (Bolnuevo and Puerto de Mazarrón) were characterized by their economic dependence on the sea. Thus, these places have acquired differentiating nuances that act as heritage identity markers that come to characterize them.

Cultural heritage is everything that is socially considered worth preserving regardless of its utilitarian interest. That is to say, it has a subjective political character, since the political powers are the first agent of activation of the heritage [35]. When researching fishing activity, one can understand the existence of a heritage that is surprising and that, in general terms, goes unnoticed by the society that is unaffiliated with fishing, and not even known, let alone valued. There is no doubt that the interest in cultural heritage in recent years has grown substantially, although it has been carried out in a heterogeneous way, as not all areas have received the same importance. Heritage linked to fishing, despite the existence of some initiatives such as in El Rompido (Huelva) [36] or in various coastal municipalities in Galicia [37], is one of the areas that has been relegated to the background in Spain, perhaps because this activity is only seen as a job, which is also extremely difficult, and inland citizens do not have a strong awareness of what this activity represents. Moreover, it is one of the last to be incorporated into heritage studies [38].

Perhaps the reason why fishing and, therefore, the heritage related to this activity goes unnoticed by society lies in the fact that citizens living inland have traditionally lived "with their backs turned" to the sea. In other words, it is an activity that takes place in a specific environment, giving rise to communities that are differentiated from those dedicated to other productive sectors [26]. This fact leads to the development of a huge fishing cultural complex, which, as in other parts of the planet, has been specialized in the sea and undervalued by the land-based world.

Fishing cultural heritage can be defined as the specific modes of material existence and social organization of the groups that make interaction with the sea their main way of life, and which include their material culture, their knowledge, their forms of social organization, their ways of interacting with nature, as well as their representation of the world, and more generally, those elements that form the basis of the identity of each group, and all of this always considered from the perspective of the productive or subsistence activity that they carry out [39].

However, fishing heritage is dynamic and changing, a result of a process of social construction that is created from the needs of the present. In addition, it is necessary to differentiate three concepts such as identity, culture and heritage due to the currently existing patrimonialization current in which the fishing activities has not been left out [40]. Fishing is not just obtaining fish for food, but it is a way of understanding life, social relationships, beliefs, landscapes and food. That is to say, there is a fishing culture in which certain elements of it can be considered as heritage and which in turn are elements that symbolically identify a people, the so called identity markers [41]. Identity, heritage and culture are closely related [42]. The activation of the maritime heritage and its recovery arise as a social necessity: to preserve the identity and uniqueness of the place and preserve the traces of an identity forgotten after decades of intensive tourism and recover the maritime past that is the origin and essence of coastal populations [43].

In this sense, fishing heritage includes identity markers common to all seafaring peoples, as well as markers specific to each people. Therefore, heritage markers have an intertwined effect on people's lifestyles and daily activities, regardless of their heritage, and they, in themselves, can become not only cultural but also economic resources [3].

In the Mazarrón Bay, there are numerous heritage and historical elements that make up the cultural fishing tradition. Thus, when travelling along the coast, three basic forms of

fishing can be observed, in accordance with [39]. Firstly, a material culture that is, with all its fishing gear and instruments, its boats, its auxiliary elements that make up the material basis of the group's activity, whether past or present. Secondly, in behavior and of learned and transmitted knowledge, in other words, everything that fishermen do to carry out their activity, to relate to each other and to interact with nature. Finally, fishing can and must also be considered from a strictly ideal perspective in the sense of ideology, that is, as the representations, ideas or images that fishermen make of themselves, of their activity, their relationship with other groups or with natural or supernatural forces; in short, their way of seeing the world, their world view.

In this sense, the fishing heritage of the Mazarrón Bay is a continuous source of cultural wealth, where we can highlight coastal defence towers built between the 15th and 16th centuries to protect the population from Berber raids [44], religious festivities related to fishing such as the Pilgrimage of Bolnuevo or the festivity of the Virgen del Carmen, whose image is carried by fishermen in the procession and even in the first festivity where sea shanties to the sea and fishing are sung [29], numerous typical dishes based on seafood products, especially tuna and shellfish, which generate a great gastronomic wealth, typical fishermen's houses, tacit knowledge about the fisherman's trade, traditional fishing gear and systems used for fishing, artisanal net repair work (Figure 2) or carpentry on the shore at the dry dock in Mazarrón Bay.

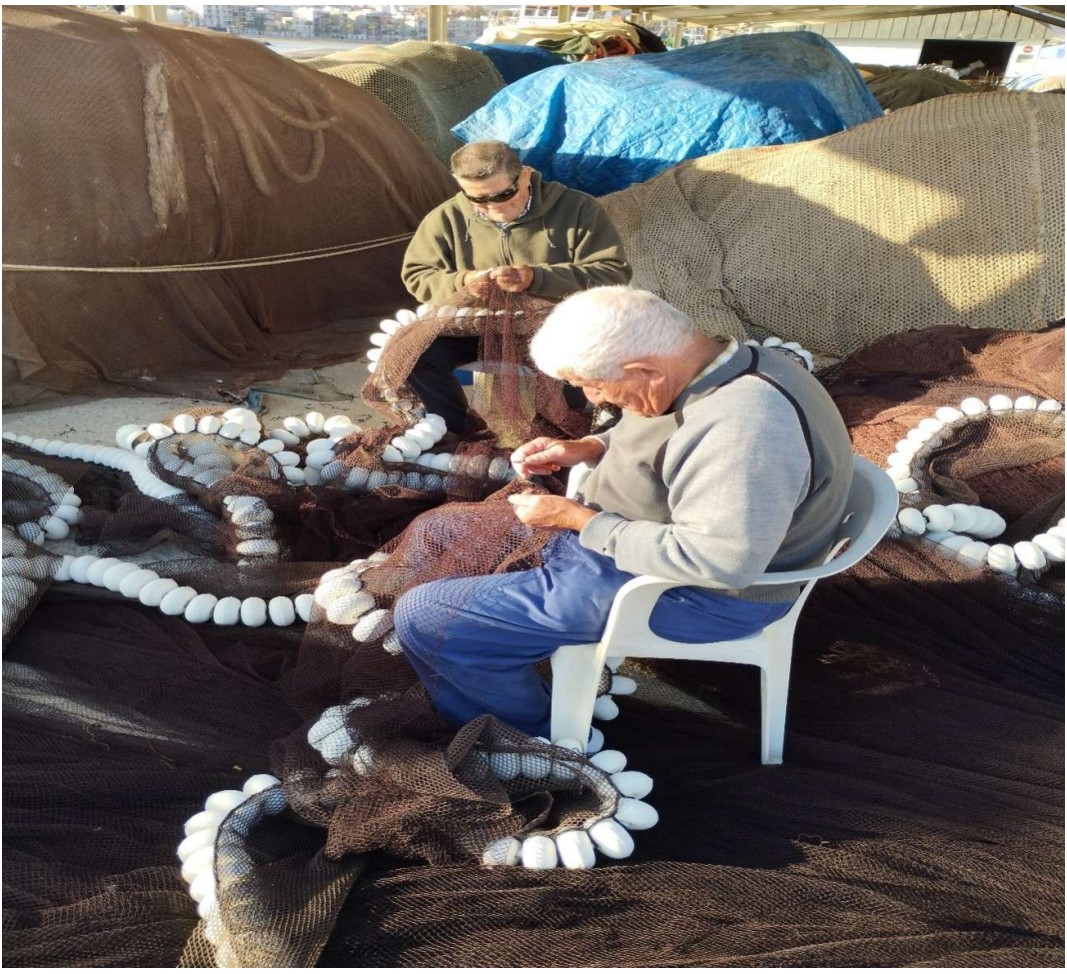

**Figure 2.** Retired fishermen repairing fishing nets today. Source: Authors.

Until several decades ago, it was possible to see fishermen in the Mazarrón Bay fishing with traditional fishing gear while these coexisted with the more modern ones. However, as is happening in other territories such as the Região Centro Norte in Portugal [45] or in

Sicily (Italy) [6], we are witnessing the extinction of an essential part of this heritage so deeply rooted in the way of life of the Mazarrón coast, with no alarm bells ringing.

## 4. The Use of Maritime Heritage

The fishing gear and systems that were used in the Mazarrón Bay during the first half of the 20th century can be considered material and intangible heritage, in the sense that they are devices for fishing and their use and employment have been passed down orally from generation to generation, being at the time a clear manifestation of tacit knowledge that, since it has not been used for decades, has been disappearing. Thus, the fact that they are no longer used for fishing and that fishing activity has been relegated to the detriment of other economic sectors such as tourism mean a loss in the collective memory of the population of these fishing systems. Thus, the holders of this knowledge, the older fishermen, are fundamental pillars for the dissemination of this heritage, their testimony being inescapable when it comes to preserving it. If they disappear, history disappears too so their safeguarding is paramount.

It is important to consider what can measures can be carried out to protect the heritage of the fishing gear used in the Mazarrón Bay, which is being forgotten. In 2003, UNESCO approved the Convention for the Safeguarding of Intangible Cultural Heritage, which defines this as "the practices, representations, expressions, knowledge and skills—together with the instruments, objects, artefacts and cultural spaces associated therewith—that communities, groups and, in some cases, individuals recognise as part of their cultural heritage". This is a definition, where the different fishing gear such as the *sardinal*, the *boguera*, the *andana* or the *boliche* (to be described in a later section) can fit in perfectly.

The importance of fishing heritage as an identity reference point results in its need to be protected, among other forms, through its documentation, registration and, fundamentally, its dissemination [41]. Nevertheless, the objectives of heritage policies are varied. The modern notion of patrimony was born associated with a role of state guardianship-protection over the nation's heritage but the current processes of patrimonialization or activation and enhancement of heritage value [46] are characterized by a markedly post-developmental and commercial orientation [47] that adopts a kind of cultural economy as a strategy for development and economic growth. In this sense, in many cases, economic interests have prevailed [48], since turning cultural resources into products continues to be a priority objective of heritage policies [49]. However, not all heritage policies are oriented towards tourists and visitors as it depends on the agent that carries it out [50]. The private sector seeks the greatest economic benefit, while the public sector carries out heritage policies that have a direct and indirect impact on the local on the local population [51]. Further, socio-community-oriented initiatives that reflect the commitment of civil society in the face of the recovery, enhancement and restitution of heritage from alternative models of management, conservation and dissemination are more frequent [52].

The fishing sector is dynamic [53]. Thus, applying policies to the heritage elements linked to fishing can be performed in two different ways in line with what was proposed by [40]. On the one hand, through documentation policies that help record the history and social evolution of the sector, a fact that is carried out in this article for the fishing gear used by the fishermen of the Mazarrón Bay during the first half of the 20th century. That is, looking to the past. On the other hand, by taking into account the past, present and future, raising awareness in society, which can become a suitable cultural and economic resource, mainly by supporting and relying on tourism [41].

In the Region of Murcia, heritage is protected by Law 4/2007, of 16 March, on Cultural Heritage of the Autonomous Community of the Region of Murcia, which states that "the cultural heritage of the Region of Murcia is made up of movable, immovable and intangible assets, such as institutions, activities, practices, uses, customs, behaviors, knowledge and manifestations of traditional life that constitute relevant forms of expression of the culture of the Region of Murcia, knowledge and manifestations of traditional life that constitute relevant forms of expression of the culture of the Region of Murcia which, regardless

of their public or private ownership, or any other circumstance that affects their legal status, deserve special protection for their enjoyment by present and future generations due to their historical, artistic, archaeological, paleontological, ethnographic, technical or industrial value or any other cultural nature".

Fishing has a long and extensive tradition in the Region of Murcia [54]. In spite of this, there is no heritage element related to fishing that has been protected under any figure of protection. The lack of protection of fishing heritage seems to be a constant trend worldwide, as the presence of intangible heritage related to fishing in UNESCO lists is very scarce, with some manifestations in Mali with the Sanké Mon (collective rite of the fishermen), in Iran with the lenj boat building techniques and in Belgium with shrimp fishing on horseback in the town of Oostduinkerke. Furthermore, few examples can be found in Spain [3]. It is worth highlighting the riverside carpentry in Coria del Río (Seville) or in Pedregalejo (Málaga), which form part of the General Catalogue of Andalusian Historical Heritage as Activities of Ethnological Interest [55], the artisanal fishing and lateen sailing in the Albufera of Valencia, or the fishing activity in the salt flats of Santa Pola (Alicante), although the latter is specifically confined to a protected natural space.

In Spain, numerous projects have been carried out in order to patrimonialize the maritime fishing culture, but they have been developed with different intensity in the autonomous coastal communities [56], being Galicia, Basque Country and Catalonia those with highest degree of development [57]. Most patrimonialization actions occur in the field of intangible heritage but attention has also been paid to tangible heritage [43].

In Catalonia, the authors of [58] have registered more than 700 actions to patrimonialize maritime culture that range from small festive events to architectural interventions, including gastronomy or the recreation of traditional boats. One of the most prominent examples is the transformation of "low-cost fish" into heritage gastronomy at Palamós Fish Space, a process involving various logics that are bringing about a change in the concept of maritime heritage [59]. In other words, it is not just a question of recovering the heritage value of fish species, fishing gear and gastronomy, but of creating a set of new heritage elements, which did not previously exist as heritage but as the culture of fishermen and their families [40]. In addition, in Catalonia important actions have been carried out in lighthouses such as Vilanova i la Geltrú, where its facilities have been converted into a museum of the sea [60] and marine catastrophes have been commemorated on various beaches so that the memory of the local fishermen continue to survive [61].

In Galicia, part of the actions to patrimonialize fishing have been carried out through associations whose objective has been to value the maritime heritage through the creation of numerous museums of fishing and the sea where aspects such as the history of this trade, traditional arts, riverside carpentry, objects related to fishermen and testimonies [62]. In addition, routes are offered that combine fishing tourism and shellfish activities with a final gastronomic tasting [63]. For their part, other Spanish regions such as Andalusia, Asturias, Cantabrian, the Valencian Community or the Murcia Region are also carrying out similar actions but with a lower degree of intensity. In this sense, there are museum spaces dedicated to fishing activity [21], being a clear example of patrimonialization of the same.

The fishing gear used by the fishermen of the Mazarrón Bay in the first half of the 20th century is a tangible and intangible heritage which is in danger of disappearing, and requires measures to safeguard it and whose continuity is at risk. The Law of Cultural Heritage of the Autonomous Community of the Region of Murcia, in Chapter V referring to ethnological heritage, considers that "the intangible assets of ethnographic value of the Region of Murcia that are in foreseeable danger of disappearance, loss or deterioration, the general directorate with competences in matters of cultural heritage will promote and adopt the appropriate measures leading to their protection, conservation, study, scientific documentation, valuation and revitalisation and to their collection by any means that guarantees their protection and their transmission to future generations". Thus, this study can be a preliminary step to urge the competent bodies to preserve this fishing heritage, writing down the composition of the different fishing gear, how the fishermen used to fish

with it, the places where they did it and their opinion on its possible use for tourism, a testimony that can be decisive in the future for its safeguarding, as fewer and fewer people who have fished with the fishing gear described in this research are living, because they are over 80 years old.

## 5. Results

### *5.1. Drifting Gear*

They can be defined as gillnets made up of rectangular pieces of net that are set in the sea without anchoring them, leaving them at the mercy of the currents, held upright by corks as floats [64,65].

### 5.1.1. *Sardinal*

*Sardinal* is fishing gear that was widely used along the Spanish coast throughout the 20th century [66], especially on the Mediterranean coast and in Galicia. As a matter of fact, today, fishermen can still be seen fishing with this method on the Galician coast, specifically in the Arousa and Muros and Noia estuaries, where the *sardinal* is called xeito [67].

The *sardinal* is made up of several rectangular pieces of netting made with vegetable fibers joined together at the ends, with one of the ends of the gear being attached to the vessel by means of a rope of variable length [68]. Each panel is approximately 100 m long and between 20 and 30 m high. In the upper rope, the corks are placed to keep the gear afloat and in the lower one the weights are placed so that it remains vertically in the sea as a barrier to ensure that sardines or other species are trapped in it. Respondents revealed that the largest *sardinals* in the Mazarrón Bay could have up to 6 panels of netting (Figure 3).

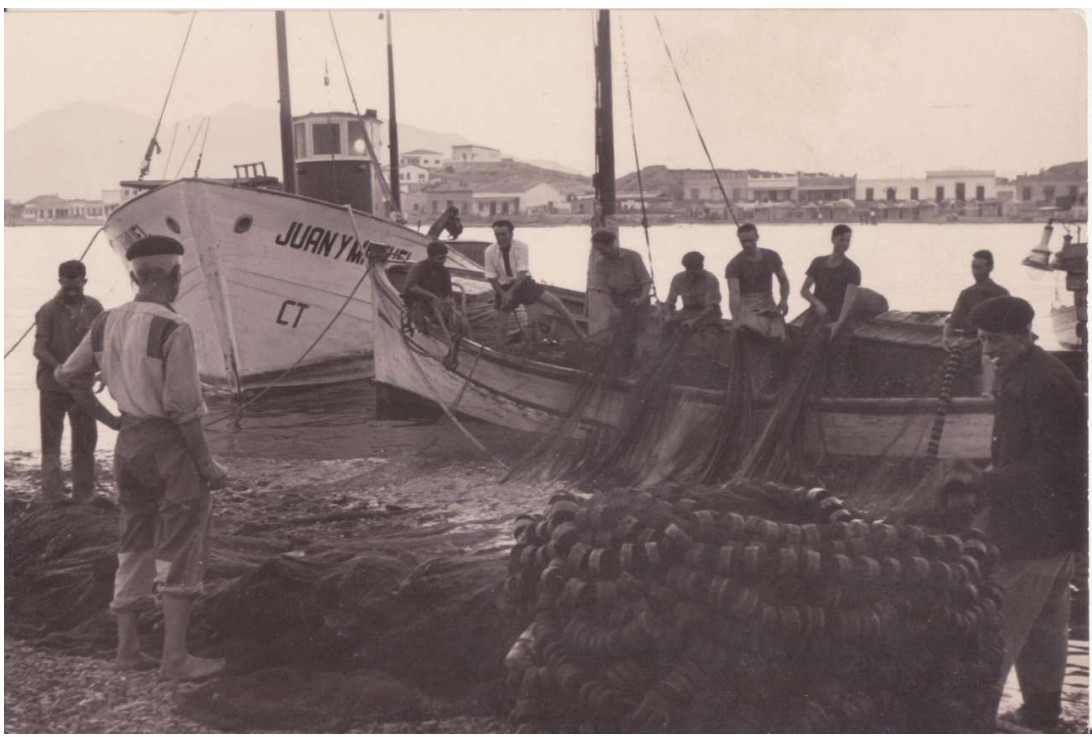

**Figure 3.** Fishermen taking fishing gear *sardinal* to the boat in the 1950s. In the background the first tourist dwellings in the town. Source: José Muñoz Muñoz.

In Mazarrón Bay, the *sardinals* were put out at dusk, perpendicular to the coastline, with the waters off the west coast of Bolnuevo and those of the Fondeadero de la Subida in Puerto de Mazarrón being the most favorable for this. The retired fishermen have revealed that throughout the night they would set the *sardinal* net several times in different places along the coast of Mazarrón in order to obtain the greatest number of catches possible, with

the summer months being the most favorable for its use. This fishing system was used to catch sardines (*Sardina pilchardus*), but as the respondents stated, they could also fish for other species such as anchovies (*Engraulis encrasicolus*), chub mackerel (*Scomber japonicus*) or Atlantic horse mackerel (*Trachurus trachurus*).

The interviewees agree on the need to enhance the value of this fishing gear, as it was one of the most commonly used by fishermen and forms part of the history of the municipality. However, due to the impossibility of fishing with it by current normative, they consider that other actions should be undertaken to make it known.

### 5.1.2. *Golondrinera*

The *golondrinera* is a fishing system that was used to catch flying fish (Cheilopogon heterurus) during the summer season, specifically during the months of June and September, when this species returns to other latitudes. The fishermen fished with this gear in the westernmost part of the Bay. That is, in the area between Bolnuevo and Cabo Cope.

The fishing gear is made up of several pieces of netting of hemp yarnsewn together with a total length of between 100 and 150 m, with a height of approximately 10 cm. As in most fishing systems, the upper rope carries a series of corks to keep the gear afloat, while the lower rope carries weights. One end is connected to the boat by a rope, while the other end is not anchored so that the *golondrinera* and the boat can move with the force of the sea currents.

The fishermen fished with the *golondrinera* in the morning, setting the gear perpendicular to the coastline and always making sure that it was in line with the currents. The collection of this gear was carried out by one fisherman at the stern of the boat, while another was collected and classified the catches. Because of this, little manpower was needed to be able to fish with the *golondrinera*.

Despite the fact that fishing with this gear would coincide with the months of greatest tourist influx in the locality, those surveyed affirmed that it should not be valued from a tourist point of view, since flying fish are no longer so frequent in the waters of the Mazarrón Bay, but from a heritage point of view.

### 5.1.3. *Lachera*

This fishing gear is similar to the *sardinal*, the only difference being that the *lachera* is smaller (15 m high). With it, fishermen tried to catch round sardinella (*Sardinella aurita*) anywhere in the Bay of Mazarrón, but also other species. Nevertheless, it was not widely used among the collective, who preferred to fish with other systems with which they could get bigger catches.

The retired fishermen acknowledge that the *lachera* could not be a tourist resource, as it is very unknown fishing gear of which there are hardly any references. Thus, several of them have affirmed that this modality disappeared after the end of the Spanish Civil War in 1939. That is to say, a little over eighty years ago, which means a great lack of knowledge about it from Mazarrón locals and even for the active fishermen themselves.

### 5.1.4. *Bonitolera*

*Bonitolera*, also called *bonitera* in some places close to the Mazarrón Bay such as in the province of Almería [60], is a fishing system dedicated especially to fishing tunas such as bonito (*Thunnus alalunga*) or frigate tuna (*Auxis thazard*).

Those surveyed affirm that the *bonitolera* is made up of several pieces of netting of vegetable fiber materials such as hemp, between 10 and 20, approximately 100 m long by approximately 20 m high, joined at their ends, giving rise to a single piece of netting that reaches a length of approximately between 1000 and 2000 m. Similar to other fishing systems, the leads are placed in the lower rope and the corks in the upper one in order to maintain the verticality of the gear, while a buoy is placed at the ends for its surface marking and stones to fix the position of the gear.

The *bonitolera* was set at sunset and lifting it took place during the early hours of the following day, the Bolnuevo coast and the La Azohía area being the most propitious for its use. The gear was set perpendicular to the coast, since the species to be captured move forming banks parallel to the mainland. An abundant number of fisherman was not necessary for its practice (between two and three men).

With regard to its enhancement in tourism, those surveyed consider that it would lack interest because its complete realization consists of two clearly differentiated parts—one is the draft and some 12 h later the lifting takes place.

### 5.2. Seine Fishing

These are techniques that consist of enclosing the species using a net that is closed once the school of fish is inside the seine [69].

### Mamparra

Retired fishermen mention the *mamparra* as the seine gear used during the first half of the 20th century on the coasts of the Mazarrón Bay. This system is similar to the one used today by purse seiners, although it was smaller due to the shorter length of the boats.

The *mamparra* was made up of a net panel made with vegetable fiber yarn measuring between 300 and 400 m long with a height of approximately 30 m. Iron rings were placed on the upper rope and on the vertical sides of the net, which formed a purse line that, once the gear was set, helped to close the lower part of the net, thus trapping the fish.

According to the fishermen interviewed, three boats were needed for the *mamparra*, one with lights to attract the fish, one to set the gear and one to sort it. The boat that illuminated the sea was the one that looked for the shoals of fish and when it found them, it gave the order to the boat containing the *mamparra* to carry out the seine. Once the encirclement had been carried out, the lower part of the gear was closed, raising it and reducing it to the smallest possible size. The catches were taken from the sea with several hooks by the crew members and thrown on board the third vessel.

The *mamparra* was used in all areas of the Mazarrón Bay to fish for pelagic species such as anchovies (*Engraulis encrasicolus*) and sardines (*Sardina pilchardus*). However, the fishermen interviewed affirmed that the Bolnuevo area was the most favorable for setting this gear. In this sense, they consider that its tourist promotion would be very positive, as it is one of the systems most used by fishermen, as it brought them abundant catches, although, as it has not been used for approximately 60 years, it is being forgotten.

### 5.3. Inshore Trawl Gear

The Inshore trawl gear consists of a large funnel-shaped mesh bag, divided into two halves, which is pulled from shore by means of ropes.

### 5.3.1. Jábega

The *jábega* is one of the most traditional fishing gears used in the coasts of Spain and Portugal for many centuries, being a system of trawling close to the coast [70,71].

The *jábega* is composed of a very elongated net of hemp yarn with a much narrower mesh in its center, called *copo* (approximately 6 mm). The sides of the net, called *pernadas*, can reach a length of 150–200 m, while the central part of the *copo* is 20–30 m long. Similar to most fishing gear, the net is attached to an upper rope with corks to keep the *jábega* wide open in the water and a lower rope with weights to keep it vertical.

The setting of the *jábega* is carried out in a group and requires great coordination between the fishermen (Figure 4).

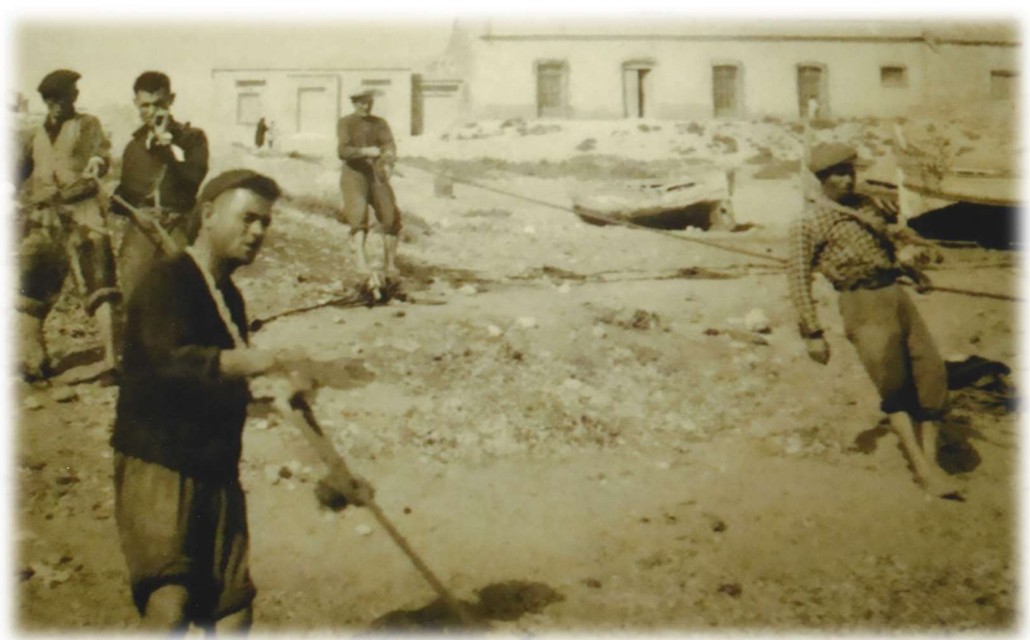

**Figure 4.** Fishermen pulling the jábega in Puerto de Mazarrón (1950s). Source: Archive of the Mazarrón Fishermen's Guild.

Some of them stay on the shore pulling the rope that holds the net to close the encirclement, while others who are in the boat row and maneuver so that the net is well deployed. Thus, to set the gear, a boat launches from the land, leaving one end of the net on the mainland, after which it sails in a straight line moving away from the coast, throwing one of the *pernadas* into the water until the *copo* sets, forming a fence. Once the *copo* is in the water, they set the other *pernadas* in the direction of land to reach it approximately 100 m from where it came out. Finally, the fishermen on the mainland, together with those who have sailed, begin to pull the gear towards the coast until the *copo* reaches the shore with the catches (Figure 4).

The practice of seine was very common in the Mazarrón Bay to capture species such as anchovies (*Engraulis encrasicolus*), sardines (*Sardina pilchardus*), sardines *(Sardinella aurita)* and whiting (*Merluccius merluccius*) until its prohibition in the early 1980s, especially in the waters of Castellar and Bolnuevo mainly, although they also settled in places such as Covaticas, Parazuelos and El Hondón.

The retired fishermen consider this fishing art as interesting tourist value. The reason is simple: for them it is one of the most emblematic, since in the past its use provided the families of the local fishermen with food, as well as a source of income, especially until 1960, when the purse seine boats began to be built with engines that had a greater extraction capacity and could fish in areas further from the coast, which caused the seine to gradually disappear little by little.

### 5.3.2. *Boliche*

*Boliche* is a fishing art similar to that of the *jábega*, but smaller, being known in the Mazarrón Bay as in other places in Spain such as the coast of Almería or Malaga as the "little brother of the *jábega*" [72].

Its method of use is similar to that of the *jábega* (Figure 5). The gear consists of a very dense net of hemp yarn that is set from land using two boats, while the rest of the fishermen wait on dry land and once the gear is finished, they pull it to the surface.

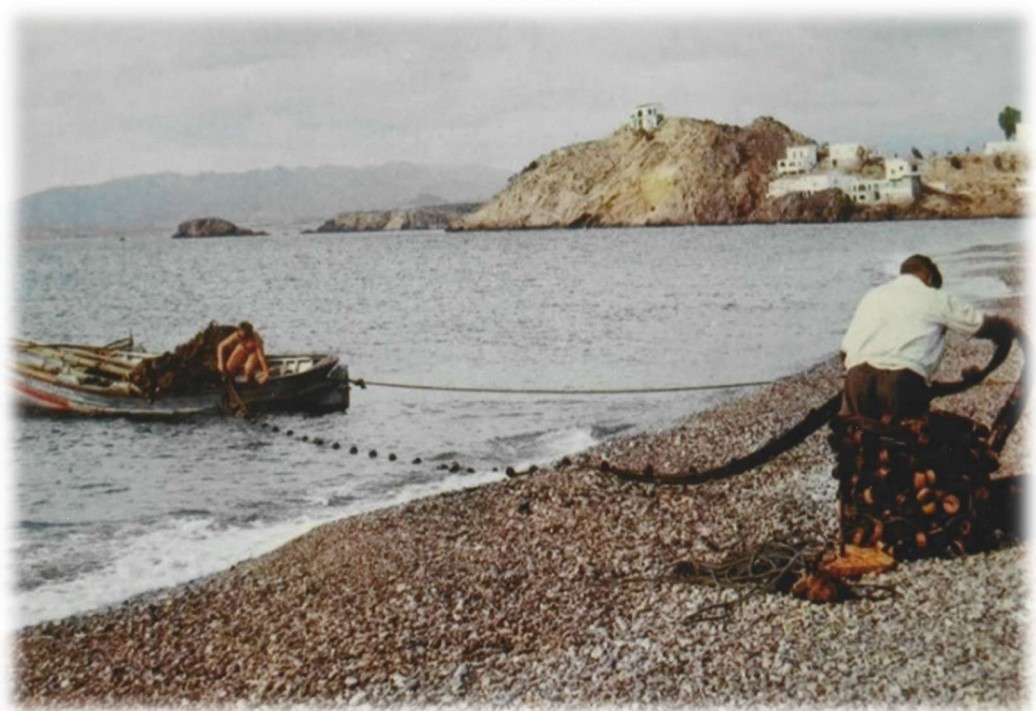

**Figure 5.** Fishermen preparing boliche in Bolnuevo (1970s). Source: Archive of the Mazarrón Fishermen's Guild.

Until the end of the 1970s, *boliche* was one of the most widely used fishing systems in the area, especially in places such as El Castellar and Bolnuevo, with fishermen catching numerous pelagic species such as anchovies (*Engraulis encrasicolus*), sardines (*Sardine pilchardus*) or Atlantic horse mackerel (*Trachurus trachurus*).

Retired fishermen consider *boliche* valuable and that it could be shown to tourists and to the citizens of the municipality in the summer, since its practice has been part of the identity of the fishermen's group for many years.

### 5.4. Seabed Fishing

These systems in which the fishing gear remains fixed to the seabed for a certain period of time before being collected by the fishermen are especially dedicated to the capture of benthic species [73,74].

#### 5.4.1. Trammel Net

The trammel net or art of three fabrics, along with *nasas*, is one of the traditional fishing gear that is still used in the Mazarrón Bay. It is a gill system fixed to the bottom of the sea that has a rectangular shape made up of three superimposed vertical netting panels made of hemp yarn, joined together by cork and lead ropes, counting the outer nets (called "little sisters" by Mazarrón fishermen given their similarity) with much larger holes than the inner net (approximately 12 cm in diameter), having a fairly large mesh size and are fully stretched in the upper rope, while the central net is much more dense (approximately 40 mm), being at the same time larger than the external ones, so that when it catches on the ropes and is looser than these, it forms pockets in which the fish are trapped when entering the gear [75]. They end up forming a crow's foot to which a rope is tied that has a stone or a cement block at one end and a piece of cork or any plastic object that stays afloat at the other end. The total length of the gear exceeds 100 m and the height is approximately 3 m, its position indicated at the ends through buoys. Nevertheless, the current trammel nets are made of synthetic materials such as polyamide.

The trammel net was a key piece of fishing gear in the first half of the 20th century for Mazarrón fishermen, given the catches it provided. It was set near the coast at sunset and pulled out in the morning, left in the sea for approximately 12 h. Two people are needed for the drafting process and subsequent lifting of the gear. In this last phase, one on the side takes the nets out of the water and another on the deck unravelling and classifying the catches. In addition, it is worth mentioning that it is still used today by small boats in the town, following the same draft process as decades ago.

The surveyed fishermen who currently fish with the trammel net affirm that they fish in the Ensenada de Mazarrón or in front of the Bolnuevo coast, some places that the older interviewees have corroborated as propitious for the practice of this art and that they too used decades ago. In other words, the fishing area for trammel nets has not changed over time and, in addition, they coincide in pointing out the variety of species caught of mollusks, crustaceans and fish such as cuttlefish (*Sepia officinalis*), octopus (*Octopus vulgaris*), conger eels (*Conger conger*), lobster (*Homarus gammarus*) or red mullet (*Mullus surmuletus*).

Trammel netting is an art that, according to currently active fishermen, could have special relevance from a tourist point of view. As they indicate, in case fishing tourism is carried out, it would be interesting to offer the visualization of the lifting of the nets due to its spectacular nature. But, for the older population surveyed, and in contrast to other traditional fishing gear, the trammel net should not be a claim, since it is a modality that is currently applied and it is not convenient to hinder the fishing task with the presence of tourists.

### 5.4.2. *Nasas*

*Nasas* are a passive fishing system that act as a trap that allows the species they are trying to catch to enter by attracting them through a bait, making it difficult for them to exit [67]. It consists of a frame or skeleton, which is what gives it shape, and a coating that covers the frame [76] of rush reeds. The entrance or funnel regulates the maximum size of the prey that enters while the lining mesh regulates the minimum size retained [77].

*Nasas* are traditionally used to capture fish, crustaceans and cephalopod mollusks such as lobsters (*Palinurus elephas*), octopus (*Octopus vulgaris*), bogue (*Boops boops*), conger eels (*Conger conger*) or moray eels (*Muraena helena*).

In the Mazarrón Bay, they were used to fish various types of *nasas*. The most frequent is the so-called simple one, which consists of a circumference of approximately two meters and approximately sixty centimeters in height. Another trap used is called "*doblera*" whose circumference is similar, but with a greater height. Finally, the "*morené*" type *nasa* (Figure 6), with a circumference of one meter and a height of half a meter.

For the *nasas* were set at dawn, all of them are pulled up at sunset of the same day at once. In the Mazarrón Bay, the practice of this art used to be performed preferably in the summer season, although it was possible to see fishermen go fishing with it during the winter. The interviewees affirm that between 40 and 80 *nasas* were set in rows, leaving a space between them of approximately 40 m, on sandy bottoms with depths between 50 and 150 m, but relatively close to rocky areas such as the island of Cueva Lobos, Cabo Cope, Puntas de Calnegre or Puntabela. Each of them is tied to a rope that has a cork at its end to keep track of its location on the surface, while stones were attached to the *nasas* so that they remained lying at the bottom of the sea.

Currently, small gear fishermen fish with *nasas*, but this activity is limited to two months a year, specifically in May and June. The interviewees affirm that the collection of the traps is interesting from the tourist point of view, since it allows observing the capture of octopus (*Octopus vulgaris*), moray eels (*Muraena helena*), squid (*Loligo vulgaris*) and lobsters (*Palinurus elephas*).

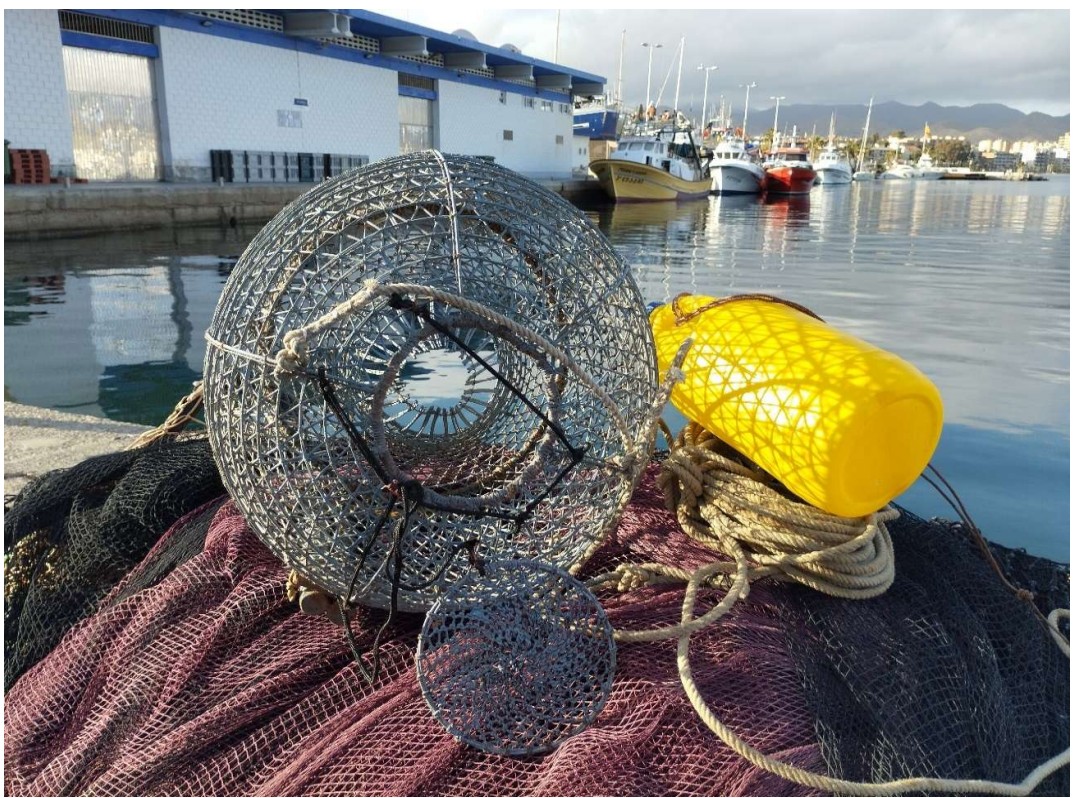

**Figure 6.** "*Morené*" type *nasa* made in the 1950s in the Mazarrón Bay. Source: Authors.

Nevertheless, they consider it to be carried out at a time when the tourist influx is not so high in the municipality. In addition, visitors usually spend the night on weekends, days when there is no fishing.

For their part, the older interviewees agree with current fishermen that this modality of fishing could be valued from the tourist point of view and make it known among its citizens.

### 5.4.3. *Andana*

*Andana* is a fishing art that can be considered as a variant of the *boguera* [68]. However, there is a clear characteristic that differentiates them, the height. Thus, the net of the *andana* measures twice that of the *boguera*. This method was also used for fishing bogue (*Boops boops*).

The older interviewees mention that the *andana* did not have any type of temporary restriction, being able to permeate all year round and be practiced in any area of the Mazarrón coastline. But, they warn that it was secondary fishing gear and if a boat set another fishing system such as a *sardinal* or *bonitolera*, the *andana* had to be set in a remote area so as not to hinder the fishing work of other boats.

Similar to the *boguera*, the interviewees consider that, since it has the same characteristics and, probably, there is currently no art of this style to practice it, the *andana* should not be classified as a resource for visitors in the summer season.

### 5.4.4. *Boguera*

The *boguera* is bottom gear that was used in the Mazarrón Bay to capture benthic species (those that inhabit the seabed) such as bogue (*Boops boops*), sea bream (*Paguellus bogaraveo*) or mackerel (*Scomber scombrus*).

The *boguera*, according to the interviewed fishermen, was set on sandy bottoms throughout the Mazarrón Bay, the boat staying very close to the coast (approximately 200 or 300 m). The art is made up of several sections of rectangular nets made with veg-

etable fibers, normally 5, which are joined at the ends, each section having a length of between 100 and 150 m. In the upper rope the corks are placed at a distance of approximately 40 cm from each other, while in the lower headlines are placed to fix the gear to the seabed. In addition, a plastic bottle is placed at each upper end of the *boguera* to track its location, as well as some stones that help to fix it so that its placement hardly varies depending on possible marine currents or the intensity of the wind.

The fishermen set the *boguera* at sunset from the stern of the boat, advancing very slowly to avoid possible entanglements in the nets. They are pulled out at dawn the following day, also from the stern, beginning with the last piece of cloth.

Retired fishermen reveal that the *boguera* ceased to be used in approximately 1960, given the existence of other much more profitable fishing systems. For this reason, they consider that valuing it can be difficult because there may not be any *boguera* in Mazarrón, although they affirm that its manufacture can be carried out without any type of problem at present, and that it should be known from the patrimonial point of view, since it has formed part of the way of life of the fishermen of the town.

## 6. Conclusions

The transformations experienced by the towns of the Spanish coast over in recent decades have had an impact on fishing activity. The fishing territories have continuously changed their way of life, diversifying their economy and betting on other production sectors such as construction and tourism, which are much more profitable. This has caused the fishing activity to remain in a less relevant position, even in some places almost marginal. However, there are many manifestations, techniques and traditional knowledge that are maintained and preserved as identifying elements of the existing social cohesion in fishing communities [78].

There is no doubt that the fishing activity has left an important and valuable legacy, from the patrimonial point of view, due to the age of its development and the variety of fishing systems used. The recovery and enhancement, as well as the subsequent maintenance of the existing elements are objectives that should be valued by the different local governments and the local populations. The reason is simple—traditionally, and in general, societies have lived with their backs to the sea, except for those that live in coastal environments. Ultimately, the culture of fishing has gone and goes unnoticed. For this reason, in many places, it is becoming extinct due to the development of mass-oriented tourism, losing heritage elements that once played a decisive role for the societies of these areas.

The fishing gear used by fishermen for decades is a good example of tangible and intangible heritage that can be forgotten. These are techniques that are practically no longer used on a day-to-day basis, and may cease to be part of the heritage in a few years.

Carrying out this study allows the preservation, in written form, of the numerous kinds of fishing gear that were used in Mazarrón Bay during the first half of the 20th century. This is fundamental for future generations, since people who have worked with these systems are of advanced age and by the time they die, they will no longer be able to be collected, hence the importance of this research to safeguard this heritage. In this sense, as occurs with tuna fishing on the coast of Cádiz, it is necessary to recognize the importance of these elements as markers of the collective being of a social group that is part of the history of the town, not sufficiently reconstructed until the moment [79]. Likewise, the methodology of this research could be applied to other maritime territories where ancient fishing gear and systems are probably being forgotten.

On the other hand, the fishermen interviewed agree on the need to publicize everything that fishing cultural heritage entails, including fishing gear. They consider it to be necessary in order to preserve the local identity of the population. Many of them affirm that various techniques such as the *jábega* could be valued in the Mazarrón Bay; as a result, we could follow the example of other places such as the Região Centro Norte in Portugal, in El Palo (Málaga, Andalusia), or in El Portús (Cartagena, Region of Murcia), where simulations

of the fishing gear are carried out in order to generate tourist visits, in which dozens of people participate.

Lastly, tourist activities in their cultural aspect require the recovery and enhancement of heritage, so that fishing cannot remain unaffected by this situation. Nevertheless, for this, it is necessary to carry out comprehensive studies that holistically group all the fishing heritage and strategies can be created to promote this in a sustainable way.

**Author Contributions:** Conceptualization, D.M.-M.; methodology, D.M.-M. and C.E.-M.; investigation, D.M.-M. and R.G.-M.; resources, D.M.-M.; data curation, D.M.-M., R.G.-M. and C.E.-M.; writing—original draft preparation, D.M.-M. and R.G.-M.; writing—review and editing, D.M.-M., R.G.-M. and C.E.-M.; visualization, D.M.-M. and R.G.-M.; supervision, R.G.-M. and D.M.-M. All authors have read and agreed to the published version of the manuscript.

**Funding:** This research received no external funding.

**Data Availability Statement:** Data sharing is not applicable for this project.

**Acknowledgments:** The authors thank the Mazarrón Fishermen's Guild. Likewise, we thank the reviewers of this paper for their timely and valuable comments and suggestions to improve the text.

**Conflicts of Interest:** The authors declare no conflict of interest.

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
