# Peer review of "Extractive Fishing Gear in the Mazarrón Bay (Murcia Region, Spain) during the First Half of the 20th Century: A Heritage Prone to Being Forgotten"

_heritage, doi:10.3390/heritage6060243_

Round 1
Reviewer 1 Report
You can find the comments in the attached file.

Author Response
Dear Reviewer:
We are very grateful for your suggestions for improvement. The paper has improved substantially. We send our comments attached.
Best regards,
Authors

Reviewer 2 Report
An interesting and valuable effort to document elements of a fishing culture that are now mostly only heritage. However, the paper can be improved.
The introductory part of this paper is too long, with much repetition of general points (i.e., what fishing heritage is and that it is important). As far as I can tell, the general idea of the introduction is that in this region of Spain, there is a long and interesting fishing tradition and heritage but it has been neglected and marginalized. This study, combining archival resources and interviews, is an effort to document that heritage in ways that can be useful to the local communities and even to the tourism that has, in an important way, sidelined the economic value of fishing to the coastal communities. All of this could be depicted rather briefly, and then move directly into the substance of the matter: the nature of the region's fisheries in the past and present, and the specific tangible and intangible heritage elements of interest.
Another problem is that the introduction provides long lists of the elements of a fisheries heritage that might be included, but the study itself reports on little more than the fishing gear and techniques used to deploy it. What about festivals? local vocabulary, stories, etc.? The introduction is written as if these matters would be included in the study, but they are not. Therefore, the introduction should be more precise about what the reader should expect.
Note also that there are many references to "the fishing heritage" of the case stud, but the fishing techniques and gear, the major focus of the heritage study, are not described at all until very late in the paper. Please try to give one or two examples of the boats or gear of whatever earlier in the paper, so that the reader can come to agreement that there is something important to this heritage.
The focus on whether and how specific techniques might be incorporated into fishery tourism is interesting and perhaps deserves more emphasis.
The quality of English is quite good. There are some small grammatical or typing errors.
I noted some small problems in writing:
39: “wich” > “which”
94 vouring its conservation [29], and these are innovative proposals that respond to the need
vouring its conservation [29], and these are innovative proposals that respond to the need
96 Moreover, it can increase the added value of fishing heritage
101 In this sense, the proliferation of fishing tourism activities is linked to Europea
126 “costal” > “coastal”
159 Like all the methodology carried out encompasses several phases.
No indication of what the traditional fishing gear or named does not appear until 263-4. This could be provided earlier.
263-4: " the different fishing gear such as the sardinal, the boguera, the andana or 263 the boliche (to be described in a later section)"
278 “The of lack of protection …” (“of”?)
340-41 Why/how impossible? "However, due to the impossibility of fishing with it, they consider that 340 other actions should be undertaken to make it known. 341"
Author Response

(The authors gave the same response as above.)

Round 2
Reviewer 2 Report
I appreciate the authors’ efforts to respond to some of the criticisms I made but feel that more needs to be done to more clearly place this work in the relevant context(s), especially the heritage policy and movements in the EU and Spain. For example, in this revision you mention the local FLAGs but you do not explain what they are and whether they are involved with this project.
Following are more particular comments,
p. 1
In other words, they fishing communities are places characterised by the human activities and social processes that have taken place there [14]. > “they” is ambiguous; how about “Fishing communities” instead?
p. 2
“fact that all cultural elements were, to a large extent, influenced by fishing and the sea [15].”
This is an overstatement; delete “all” and consider replacing with “many” or “some.”
“These places help to understand the culture of fishermen and the significance of heritage in their daily lives [17].” How do “places” lead to an understanding of culture and heritage?
“The concept of heritage implies that the tangible and intangible goods that form it have been inherited by a social group and preserved over time to be passed on to future generations.”
---this understanding of the concept of heritage could just as well be “tradition.” Unless you feel that “heritage” is simply that part of culture and experience that comes from the past. The other reviewer took a more critical, political perspective on “heritage,” as something that is socially constructed using traditions from the past—a more selective process than just passing on to the next generation. Indeed, this article itself is part of such a process, trying to make the older fishing gears part of today’s notion of the “heritage” of the community.
62ff: The revisions are problematic:
“nevertheles” > “nevertheless”
“Traditional fishing gear 64 and its their uses have been disappearing due to the dynamics of the sector. For example, 65 nowadays it is not fished with fishing gear such known as sardinal, andana, boliche or the jábega 66 that had a unique boat [what does this mean?] are not longer used in places like Málaga. “
“According to [19], it is necessary to enhance 67 the value of elements related to fishing activity and coastal areas, in order to prevent their 68 deterioration and subsequent disappearance of these heritage elements.”
Reference #19 is about the past and future of patrimony. It might be helpful to reflect more about this here in the introduction of the paper, to do a better job of placing this in some intellectual context. Otherwise in this statement, you are simply saying that increasing the value of the fishing gear and activities is a way to prevent their decline and disappearance. The big question is, why should anyone care about their decline and disappearance? What does this mean to the local communities? To folklorists, to local historians, to historians of maritime Spain? That’s where the construction of heritage comes in.
In the next paragraph you briefly mention the FLAGs and patrimonialization of maritime culture, and that does seem to be a big clue to the important and relevant context of this research. It could be highlighted rather than offered as an extra sentence at the end.
“In this sense, and 69 aspects such as social alterations resulting from the changing lifestyle of the population 70 [20], gentrification modifying the heritage culture and economy of port cities or the intensification of mass 71 coastal tourism [21] make the local population become more aware of heritage. “
Please clarify.
I offer a little bit of editing above, but what is the general meaning? Do you mean that the local population is more aware of the importance of ‘heritage’ of fishing (and other things) because of demographic change, gentrification, tourism. Is this research done because some of the local people asked for more information about their heritage, because they are aware that it has become less visible with these changes?
Clearly there is a movement in Spain and the EU to protect the heritage, or patrimony, of fishing in coastal communities, but that gets lost in these other discussions. So there needs to be more of an explanation of the FLAGs in the region and how this research relates to their mission.
80 and these there are innovative proposals
“Given 95 these circumstances, various organisations such as the Fishermen's Guild and the Town 96 Council are looking for ways to promote the fishing culture in the municipality…. “ What about the FLAGS mentioned earlier???
“The oral accounts of the fishermen have also been gathered in order 105 to preserve them with the possibility of making the most of them for tourism.In this way, 106 they could be occasionally recreated on the coasts of the Mazarrón Bay as a way of en- 107 hancing maritime culture.”
It is unclear what “them” refers to—the oral accounts or the fishing gear? One reading of this is that the oral accounts will be preserved, with the potential of using them and the information in them for tourism. That makes a lot of sense. Without oral accounts of the people who actually worked with the fishing gear, the heritage loses much of its value. But perhaps you only used the interviews with the elderly fishermen to learn more about the gear, so that some of them can be used or revived for tourism. This is unclear. Will a museum or other entity actually preserve those interviews for interested people to view?
“That is, the fishing culture is transformed by changing the habits of fishermen given the creation of this infrastructure.” Too vague. Changed in what ways? If you are not planning to describe this, do not say it.
:Moreover, at the same time, houses for fishermen are being built [13] in Puerto de Mazarrón, so part of Bolnuevo fishermen change their place of residence to a larger house with more amenities. “ I assume that this occurred in the early 1970s. Therefore changes to grammatical tense are required: Moreover, at the same time, houses for fishermen are being were built [13] in Puerto de Mazarrón, so part some of Bolnuevo fishermen changed their place of residence to a larger houses with more amenities. “
Lines 137-139: you state that because it was a small-scale fishery there was a lower volume of catch than now is the case. However, it is quite possible for a small-scale fishery to have a higher volume of catch in the past, depending on how many were involved and how abundant the fish were! If you do know that the volume of catch was lower in the past, just say that it was a small-scale fishery and the volume of catch was lower than it is today.
I'm afraid that I do not have time to continue going through the paper line by line. My major concern now is that you more appropriately contextualize the work you have done in relation to the movements in the EU and Spain on the patrimony of maritime communities and make clear what the FLAGs are and how they relate to your work.
By and large you have done very well using the English language. I have noted a few errors. You should ask another native English speaker to review the paper as well.
Author Response
Dear Reviewer:
We attach a letter with comments to your appropriate suggestions.
Thank you very much for the valuable advices. Truly, you are a great expert in these research topics.
Best regards,
Authors
